# Reconstruction of daily gridded snow water equivalent product for the Pan-Arctic region based on a ridge regression machine learning approach

Donghang Shao[1,2], Hongyi Li[1,2], Jian Wang[1,2], Xiaohua Hao[1,2], Tao Che[1,2] and Wenzheng Ji[1,2]

[1]Northwest Institute of Eco-Environment and Resources, Chinese Academy of Sciences, Lanzhou, 730000, China

[2]Heihe Remote Sensing Experimental Research Station, Key Laboratory of Remote Sensing of Gansu Province, Chinese Academy of Sciences, Lanzhou, 730000, China

*Correspondence to*: Hongyi Li (lihongyi@lzb.ac.cn)

**Abstract.** Snow water equivalent is an important parameter of the surface hydrological and climate systems, and it has a profound impact on Arctic amplification and climate change. However, there are great differences among existing snow water equivalent products. In the Pan-Arctic region, the existing snow water equivalent products are limited time span and limited spatial coverage, and the spatial resolution is coarse, which greatly limits the application of snow water equivalent data in cryosphere change and climate change studies. In this study, utilizing the ridge regression model (RRM) of a machine learning algorithm, we integrated various existing snow water equivalent (SWE) products to generate a spatiotemporally seamless and high-precision RRM SWE product. The results show that it is feasible to utilize a ridge regression model based on a machine learning algorithm to prepare snow water equivalent products on a global scale. We evaluated the accuracy of the RRM SWE product using Global Historical Climatology Network (GHCN) data and Russian snow survey data. The MAE, RMSE, R, and $R^2$ between the RRM SWE products and observed snow water equivalents are 0.24, 30.29 mm, 0.87, and 0.76, respectively. The accuracy of the RRM SWE dataset is improved by 24%, 25%, 32%, 7%, and 10% compared with the original AMSR-E/AMSR2 snow water equivalent dataset, ERA-Interim SWE dataset, Global Land Data Assimilation System (GLDAS) SWE dataset, GlobSnow SWE dataset, and ERA5-land SWE dataset, respectively, and it has a higher spatial resolution. The RRM SWE product production method does not rely too much on an independent snow water equivalent product, it makes full use of the advantages of each snow water equivalent dataset, and it considers the altitude factor. The average MAE of RRM SWE product at different altitude intervals is 0.24 and the average RMSE is 23.55 mm, this method has good stability, it is extremely

suitable for the production of snow datasets with large spatial scales, and it can be easily extended to the preparation of other snow datasets. The RRM SWE product is expected to provide more accurate snow water equivalent data for the hydrological model and climate model and provide data support for cryosphere change and climate change studies. The RRM SWE product is available from the 'A Big Earth Data Platform for Three Poles' (http://dx.doi.org/10.11888/Snow.tpdc.271556) (Li et al., 2021).

## 1 Introduction

The IPCC (Intergovernmental Panel on Climate Change) AR6 (Sixth Assessment Report) notes that the Northern Hemisphere spring snow cover has greatly reduced since 1950, and the feedback effect of the climate system caused by this reduction is extremely large (Masson-Delmotte et al., 2021). In most land areas of the Northern Hemisphere, annual runoff is dominated by snowmelt, and accurately estimating the impacts of such a large amount of snowmelt runoff on ecosystems and human activities is of great significance (Barnett et al., 2005; Bintanja and Andry, 2017; Henderson et al., 2018). Whether through hydrometeorological simulation or global change research, the estimation of energy budget and mass of snow is very difficult, so a set of highly accurate, long time series snow cover datasets is urgently needed to drive hydrometeorological simulations and land surface process models. Among them, snow water equivalent data play an irreplaceable role as an important parameter of the land surface hydrological model and climate model.

At present, there are many forms of snow water equivalent data in the world. According to type, these data can be divided into site observation snow water equivalent (SWE), remote sensing SWE, reanalysis SWE, data assimilation SWE and model simulation SWE. The remote sensing SWEs are mainly AMSR-E (Kelly, 2009) and AMSR2 (Imaoka et al., 2010; Tedesco and Jeyaratnam, 2019). The reanalysis SWE mainly includes ERA-Interim (Dee et al., 2011), MERRA2 (Gelaro et al., 2017), MERRA land (Reichle et al., 2011), and ERA5-land (Muñoz Sabater, 2019; Balsamo et al., 2015). The data assimilation SWE mainly includes GlobSnow (Luojus et al., 2021) and Global Land Data Assimilation System (GLDAS) (Rodell et al., 2004). The site observation SWE mainly includes the GHCN dataset (Menne et al., 2016). However, the time ranges of AMSR-E and AMSR-E2 SWE are only from 2003 to present, which is lacking in terms of time series. Similarly, the GlobSnow SWE dataset

is also seriously lacking in time series. Although the reanalysis SWE data have good spatial and temporal continuity and high data integrity, their accuracy is poor, and its MAE is 0.65 (Snauffer et al., 2016). The snow water equivalent data from stations and meteorological observations cannot meet the needs of hydrometeorological and climate change research. This is mainly because SWE from stations is discontinuous in time series and severely missing. Further, hydrometeorological studies often require spatiotemporally continuous grid data to be driven (Pan et al., 2003). There are great differences among remote sensing SWE, reanalysis SWE data, data assimilation SWE and observation SWE. For remote sensing SWE, the spatio-temporal characteristics of different passive microwave snow water equivalent data differ significantly due to differences in sensors or retrieval algorithms (Mudryk et al., 2015a). For data assimilation SWE and reanalysis SWE data, they also tend to exhibit different spatio-temporal characteristics due to differences in model design, driving data, assimilation methods, etc. (Vuyovich et al., 2014). In summary, although there are a variety of snow water equivalent data in the world, the data quality is uncertain.

Previous studies have shown that all kinds of snow water equivalent data in the Northern Hemisphere have advantages and disadvantages, and none of these data perform well in all aspects (Mortimer et al., 2020). An effective method is to fuse all kinds of snow water equivalent data in time and space, integrate the advantages of all kinds of data, and then generate a relatively complete snow water equivalent dataset. Many scholars have conducted in-depth studies on snow water equivalent data fusion. The main fusion methods can be classified into the following categories: multiproduct direct average (Mudryk et al., 2015b), linear regression (Snauffer et al., 2016), data assimilation (Pulliainen, 2006), "multiple" collocation (Pan et al., 2015) and machine learning (Snauffer et al., 2018; Xiao et al., 2018; Wang et al., 2020). Studies have shown that even the simplest multisource data average is more accurate than a single snow water equivalent product (Snauffer et al., 2018). However, the simple multisource data average cannot highlight the advantages of high-precision data, and it is easily affected by the weight ratio of low-precision data, which reduces the accuracy of fused data (Mudryk et al., 2015a). Although the linear regression method can make good use of the actual observation data to correct the original data, it is easy to overfit and causes the overall deviation (Snauffer et al., 2016). The "multiple" collocation method changes the size of the original SWE data before fusion, which easily causes data errors. The data assimilation method is sensitive to the accuracy of input data, and it is difficult to fuse multisource data (Pan et al., 2015). In recent years, machine learning methods have been widely used in data fusion (Santi et al., 2021; Ntokas et al., 2021). Machine learning methods can not only integrate the advantages of multisource

data but also make full use of site observation data to train the sample data, which easily generates snow water equivalent data products with large spatial scales and long time series (Broxton et al., 2019; Bair et al., 2018).

In summary, based on the existing snow water equivalent data products, combining a machine learning algorithm to fuse multisource snow water equivalent data is an effective method to prepare snow water equivalent products with long time series and large spatial scales and retain the advantages of single snow water equivalent data products. In this study, we integrated multisource snow water equivalent data products of RRM SWE based on the ridge regression model of the machine learning algorithm. We selected ERA-Interim SWE data, GLDAS SWE data, GlobSnow SWE data, AMSR-E/AMSR2 SWE data, and ERA5-land SWE data with relatively complete time series as the original data for the production of RRM SWE product. The missing parts of the ERA-Interim SWE data, AMSR-E/AMSR2 SWE data, and GlobSnow SWE data are filled by the spatial-temporal interpolation method. The GHCN dataset (Menne et al., 2016) and Russian snow survey data (Bulygina et al., 2011) are used as training sample data of "true snow water equivalent", and the effect of altitude on the algorithm is also considered. Thus, we prepared a set of spatiotemporal seamless snow water equivalent datasets (RRM SWE) covering the Pan-Arctic region from 1979 to 2019. The spatial coverage of the RRM SWE product covers all land regions north of 45° N.

## 2 Data and methods

### 2.1 Research region

The research region of the RRM SWE product is located in the land region north of 45° N (hereinafter referred to as the Pan-Arctic region) (Fig. 1). This region consists of Asia, Europe, and North America. The land region covers Russia, the United States, Canada, Denmark, Norway, Iceland, Sweden, and Finland. This region has a cold climate and a wide area of snow cover.

### 2.2 Grid snow water equivalent data description

In this study, we utilize ERA-Interim SWE data (Dee et al., 2011), GLDAS SWE data (Rodell et al., 2004), GlobSnow SWE data (Luojus et al., 2021), AMSR-E/AMSR2 SWE data (Tedesco and Jeyaratnam, 2019), and ERA5-land SWE data (Muñoz

Sabater, 2019) as the original input datasets for the fusion data (Table 1).

GlobSnow is a dataset of global snow cover and snow water equivalents for the Northern Hemisphere released by the European Space Agency (ESA) (http://www.globsnow.info/swe/) (Luojus et al., 2021). The SWE products in this dataset combine the Canadian Meteorological Center (CMC) daily snow depth analysis data (Walker et al., 2011), ground weather site observation data, and satellite microwave radiometer data. We obtained the L3A_daily_SWE product of this dataset. The temporal resolution of the L3A_daily_SWE product is daily, the spatial resolution is 0.25°, and the data format is NETCDF4.

ERA-Interim is the fourth generation reanalysis data of the European Centre for Medium-Range Weather Forecasts (ECMWF) (Dee et al., 2011). The data provide a global assimilated numerical product of various surface and top atmospheric parameters from January 1979 to present (https://apps.ecmwf.int/datasets/data/interim-full-daily/levtype=sfc/). We obtained the snow water equivalent dataset with a daily temporal resolution, a spatial resolution of 0.25°, and NETCDF4 data format. The spatial range of the data is the Pan-Arctic region north of 45°N.

The Advanced Microwave Scanning Radiometer-Earth Observing System (AMSR-E) is a microwave scanning radiometer on the Aqua satellite of the National Aeronautics and Space Administration (NASA) Earth Observing System (EOS) (Tedesco and Jeyaratnam, 2019). The AMSR-E provides a global daily snow water equivalent dataset from June 19, 2002, to October 3, 2011 (https://nsidc.org/data/ae_dysno). AMSR2 is a microwave scanning radiometer on the GCOM-W1 satellite launched by the Japan Aerospace Exploration Agency (JAXA) in May 2012. AMSR2 provides a global snow water equivalent dataset from July 2, 2012 to present (https://nsidc.org/data/AU_DySno/versions/1). The spatial resolution of the AMSR-E SWE and AMSR2 SWE datasets is 25 km x 25 km, the temporal resolution is daily, and the data formats are HDF-EOS and HDF-EOS5, respectively.

The GLDAS is a model used to describe global land information; it contains data, such as global rainfall, water evaporation, surface runoff, underground runoff, soil moisture, surface snow cover distribution, temperature, and heat flow distribution (Rodell et al., 2004). This assimilation system includes data with spatial resolutions of 1°×1° and 0.25°×0.25° and temporal resolutions of 3 hours, 1 day and 1 month. The GLDAS data are available for download from the Goddard Earth Sciences Data and Information Services Center (GES DISC). We obtain a snow water equivalent dataset with the daily temporal resolution, 0.25° spatial resolution, and NETCDF4 data format.

ERA5-land is a reanalysis dataset that provides the evolution of global land parameter data since 1981 (Muñoz Sabater, 2019). The dataset provides eight types of snow parameter data, including snow albedo, snow cover, snow depth, snowfall, the temperature of the snow layer, snowmelt, snow density, and snow water equivalent. This dataset provides a global snow water equivalent dataset with a spatial resolution of hourly, the temporal resolution of 0.1°×0.1°, temporal coverage of January 1981 to present, and data formats of GRIB and NETCDF4.

To maintain consistency in the spatial and temporal resolutions of the fused data, we unified the ERA-Interim SWE data, GLDAS SWE data, GlobSnow SWE data, AMSR-E/AMSR2 SWE data, and ERA5-land SWE data into a daily temporal resolution, with a spatial resolution of 0.25° and geographic projection of North Pole Lambert Azimuthal Equal Area.

### 2.3 Ridge regression machine learning algorithm for preparing snow water equivalent

In this study, we utilize the ridge regression model of a machine learning algorithm to fuse ERA-Interim SWE data (Dee et al., 2011), GLDAS SWE data (Rodell et al., 2004), GlobSnow SWE data (Luojus et al., 2021), AMSR-E/AMSR2 SWE data (Tedesco and Jeyaratnam, 2019), and ERA5-land SWE data (Muñoz Sabater, 2019) to generate a set of new snow water equivalent dataset of RRM SWE. The target reference data in this study are the GHCN dataset and Russian snow survey data.

The ridge regression model is a biased estimates regression method for collinear data analysis (Friedman et al., 2010; Hoerl and Kennard, 1970b, a). By abandoning the unbiasedness of the ordinary least squares, this algorithm can obtain the regression method in which the regression coefficient is more practical and reliable at the cost of losing part of the information and reducing the accuracy. This model has high fitting accuracy for ill-conditioned data. The advantage of this model is that it uses simple, accurate, and easy to prepare snow water equivalent products with long time series and large spatial scales. The principle equation of the ridge regression model is defined as follows:

$$\hat{\beta}^{ridge} = \underset{\beta}{\operatorname{argmin}} \left\{ \sum_{i=1}^{N} \left( y_i - \beta_0 - \sum_{j=1}^{p} x_{ij}\beta_j \right)^2 + \lambda \sum_{j=1}^{p} \beta_j^2 \right\}, \tag{1}$$

where $\hat{\beta}^{ridge}$ is the extremum solution function of ridge regression. $p$ is the number of variables involved in training. $x_i$ is the predicted snow water equivalent, $y_i$ is the observed snow water equivalent, and $\lambda$ , $\beta$ , $\beta_j$ and $\beta_0$ are the parameters

to be solved. $1,\cdots,N$ is the sample of the training dataset. $\lambda\sum_{j=1}^{p}\beta_j^2$ is the penalty function terms. The model is developed in python3, and the model framework is based on the "scikit-learn" machine learning library (https://scikit-learn.org/stable/index.html), and the code is available.

The integration process of the RRM SWE product (Fig. 2) is described as follows:

1) The original ERA-Interim SWE data, GLDAS SWE data, GlobSnow SWE data, AMSR-E/AMSR2 SWE data, ERA5-land SWE data, digital elevation model (DEM) data, unified temporal resolution, spatial resolution, projection, spatial range, and unit are preprocessed.
2) The spatiotemporal interpolation method is used to fill in the missing data of AMSR-E/AMSR2 SWE, ERA-Interim SWE, and GlobSnow SWE in space and time. Based on this method, the missing data of AMSR-E/AMSR2 SWE at low latitudes and the missing data of ERA-Interim SWE and GlobSnow SWE on time series are filled.
3) The snow water equivalent data observed at stations from 1979 to 2014 are used as sample training data, and the AMSR-E/AMSR2 SWE, ERA-Interim SWE, GLDAS SWE, GlobSnow SWE, ERA5-land SWE data, and DEM data are input into the ridge regression model of a machine learning algorithm for training. During the model training process, we restructured the training data, reduced the training data appropriately for the regions with denser training data, and make it close to the amount of training data in the sparse region.
4) When the model was trained, ERA-Interim SWE, GLDAS SWE, GlobSnow SWE, and ERA5-land SWE were used for training data between 1979 and 2002 (AMSR-E/AMSR2 SWE data were not available before 2002.), and AMSR-E/AMSR2 SWE, ERA-Interim SWE, GLDAS SWE, GlobSnow SWE, and ERA5-land SWE were used for training data after 2002.
5) Based on the S-Fold Cross Validation method, the snow water equivalent data are continuously trained and validated, and finally select the optimal model and parameters are evaluated by the loss function.
6) Based on the trained optimal model, multiple snow water equivalent data products are integrated into the time series, missing data are predicted, and a set of spatiotemporally seamless snow water equivalent datasets is generated.

7) Snow water equivalent data observed at stations from 2015 to 2019 are used to evaluate the accuracy of the RRM SWE product.

### 2.4 Site data and evaluation metrics

#### 2.4.1 Site snow water equivalent data for training, validation, and testing

Russian snow survey data (http://aisori.meteo.ru/ClimateR) include the average snow depth data and the average snow density data of the station, and the snow water equivalent is the product of the measured average snow depth and the average snow density (Bulygina et al., 2011). We obtained the snow water equivalent data of 19493 stations in 1979-2016 from this dataset.

GHCN-Daily is a comprehensive dataset that records the historical temperature, precipitation, and snow cover of the global land area (Menne et al., 2016). This dataset is from the National Oceanic and Atmospheric Administration (NOAA) (ftp://ftp.ncdc.noaa.gov/pub/data/ghcn/daily/by_year/). The dataset provides data from 75000 observation sites in 179 countries around the world. This dataset contains more than 40 meteorological elements, such as temperature, precipitation, snow depth, snow water equivalent, wind speed, and evaporation capacity. We obtained data from all sites containing snow water equivalents from 1979 to 2020.

We carefully screened the Russian snow survey data and GHCN data and eliminated some abnormal observation data to ensure the high quality of the train set, validation set, and test set. The null values, negative numbers, and extreme snow water equivalent value greater than 2000 mm are removed during the GHCN data screening process. The null values, negative numbers, and extreme snow water equivalent value greater than 2000 mm are removed during the Russian snow survey data screening process.

#### 2.4.2 Accuracy evaluation method for datasets

Mean absolute error (MAE), root mean square error (RMSE), Pearson's correlation coefficient (R), and coefficient of determination ($R^2$) are used to evaluate the accuracies of AMSR-E/AMSR2 SWE, ERA-Interim SWE, GLDAS SWE, GlobSnow SWE, ERA5-land SWE, and the RRM SWE product. The specific equation of accuracy evaluation error is described as follows.

$$MAE = \frac{1}{n}\sum_{i=1}^{n}\left| f_i - y_i \right|, \tag{2}$$

$$RMSE = \left[\frac{\sum_{i=1}^{n}(f_i - y_i)^2}{n}\right]^{\frac{1}{2}}, \tag{3}$$

$$R = \frac{1}{n-1}\sum_{i=1}^{n}\left(\frac{f_i - \overline{f}}{\sigma_f}\right)\left(\frac{y_i - \overline{y}}{\sigma_y}\right), \tag{4}$$

$$R^2 = \frac{\sum_{i=1}^{n}\left(f_i - \overline{y}\right)^2}{\sum_{i=1}^{n}\left(y_i - \overline{y}\right)^2}, \tag{5}$$

where $n$ is the sample of the validation dataset, $f_i$ is the snow water equivalent dataset product, and $y_i$ is the measured snow water equivalent at the station. $\overline{f}$ and $\overline{y}$ are the averages of snow water equivalent products and measured snow water equivalents, respectively. $\sigma_f$ and $\sigma_y$ are the standard deviation of snow water equivalent products and measured snow water equivalents, respectively.

To further evaluate the accuracy of the RRM SWE dataset at the spatial scale, we compared it with AMSR-E/AMSR2 SWE, ERA-Interim SWE, GLDAS SWE, GlobSnow SWE, and ERA5-Land SWE at different altitude gradients. We also evaluate MAE, RMSE, R and $R^2$ separately for 11 elevation intervals: <100 m, 100-200 m, 200-300 m, 300-400 m, 400-500 m, 500-600 m, 600-700 m, 700-800 m, 800-900 m, 900-1000 m, and >1000 m.

We use the Mann-Kendall trend test (Mann, 1945; Kendall, 1990) method to evaluate the variation trend in the RRM SWE dataset from 1979 to 2019 and analyze its reliability in terms of time series. Since the AMSR-E/AMSR2 SWE product and the GlobSnow SWE product lacks snow water equivalent data for Greenland, we removed Greenland data to maintain consistency in the spatial extent of the comparison data.

## 3 Results and discussion

### 3.1 Overall accuracy evaluation of the RRM SWE product

In this study, the accuracy of the RRM SWE, AMSR-E/AMSR2 SWE, ERA-Interim SWE, GLDAS SWE, GlobSnow SWE, and ERA5-land SWE was compared using test datasets from 2015 to 2019. MAE, RMSE, R, and $R^2$ were used to reflect the data quality of each snow water equivalent product.

According to the verification results in Fig. 3 and Table 2, the RRM SWE data have the best overall accuracy, and the MAE, RMSE, R, and $R^2$ between the observed snow water equivalents are 0.24, 30.29 mm, 0.87, and 0.76, respectively. The overall accuracy of the GlobSnow SWE and ERA5-land SWE products is higher than that of other snow water equivalent products. The overall deviation of the GlobSnow SWE products is the smallest except for the RRM SWE data, with MAE and RMSE values of 0.31 and 35.21 mm, respectively. Although the overall deviation between the GlobSnow SWE dataset and the measured snow water equivalent is small, its correlation with the measured value is lower than that of the ERA5-land SWE dataset, and it is missing data in terms of time series. The correlation between ERA5-land SWE and observed snow water equivalent is the highest except for the RRM SWE data, with R and $R^2$ values of 0.84 and 0.70, respectively. The overall deviation between the ERA5-land SWE dataset and the measured snow water equivalent is higher than that of the GlobSnow SWE dataset, but its correlation with the measured values was higher than that of the GlobSnow SWE dataset, and its integrity is better in terms of temporal and spatial series. In addition, the overall accuracy of the ERA-Interim SWE dataset and GLDAS SWE dataset is relatively low, but their integrities are higher than that of the GlobSnow SWE dataset and AMSR-E/AMSR2 SWE dataset in terms of temporal and spatial series. The AMSR-E/AMSR2 SWE dataset has a higher estimation accuracy for the low-value region of snow water equivalent. Moreover, in the Pan-Arctic region, most of the existing snow water equivalent data products are missing to varying temporal and spatial degrees. Obviously, the accuracies of the existing snow water equivalent products were uneven, and any kind of snow water equivalent dataset is not absolutely perfect.

The verification results also indicate the following ranking orders:

The MAE ranking order is RRM SWE< GlobSnow SWE< ERA5-land SWE< AMSR-E/AMSR2 SWE< ERA-Interim SWE< GLDAS SWE.

The RMSE ranking order is RRM SWE< GlobSnow SWE< ERA5-land SWE< AMSR-E/AMSR2 SWE< ERA-Interim SWE< GLDAS SWE.

The R ranking order is RRM SWE > ERA5-land SWE > GlobSnow SWE > ERA-Interim SWE > AMSR-E/AMSR2 SWE > GLDAS SWE.

The $R^2$ ranking order is RRM SWE > ERA5-land SWE > GlobSnow SWE > ERA-Interim SWE > AMSR-E/AMSR2 SWE > GLDAS SWE.

Compared with AMSR-E/AMSR2 SWE, ERA-Interim SWE, GLDAS SWE, GlobSnow SWE, and ERA5-land SWE, the MAE of the RRM SWE and observed snow water equivalent is reduced by 0.23, 0.24, 0.32, 0.07, and 0.09, respectively. The RMSE of the RRM SWE and observed snow water equivalent is reduced by 17.22 mm, 21.69 mm, 32.54 mm, 4.91 mm, and 7.82 mm, respectively. The correlation coefficient of the RRM SWE and observed snow water equivalent is improved by 0.30, 0.24, 0.45, 0.11, and 0.02, respectively. The coefficient of determination of the RRM SWE and observed snow water equivalent is improved by 0.45, 0.37, 0.59, 0.20, and 0.06, respectively. Based on the above verification results, the accuracy of the RRM SWE is significantly improved; the RRM SWE dataset has higher accuracy than any single grid snow water equivalent dataset, and it also fills the gap in the original snow water equivalent data in terms of spatial and temporal resolutions.

Based on the kernel density estimation method, we analyze the density distribution of different SWE datasets (Fig. 4). The results show that the RRM SWE dataset is closer to the 1:1 line and has the highest accuracy. The RRM SWE dataset is particularly accurate for SWE estimation in the low-value region, and the test data are concentrated near the 1:1 line in the high-density region (kernel density estimation > 0.00015) (Fig. 4). In contrast, the high-density regions of the GLDAS SWE dataset, ERA-Interim SWE dataset, and AMSR-E/AMSR2 SWE dataset deviate significantly from the 1:1 line, resulting in poor accuracy. The AMSR-E/AMSR2 SWE, GLDAS SWE, and GlobSnow SWE are underestimated relative to the snow water equivalent measured at the site, among which GLDAS SWE underestimated the observed snow water equivalent the most seriously, while ERA5-land SWE overestimated the observed snow water equivalent. Although the accuracies of GlobSnow SWE and ERA5-land SWE are relatively high, their dispersion degrees are large (the kernel density estimation for most test data is less than 0.0001). Overall, the RRM SWE data have a higher overall estimation accuracy, especially for the low-value area of snow water equivalent.

However, in this study, there are still some uncertainties in the ridge regression machine learning model that integrates snow water equivalent products. First, this model is strongly dependent on on-site observation data, and the fusion precision of snow water equivalent is poor in some areas with sparse observation stations. The fusion accuracy of snow water equivalent products will be affected to a certain extent without considering the prior snow cover information. Then, an underestimation of high SWE remains in the RRM SWE product. The main reason is that there are few high SWE data in GHCN data and Russian snow survey data, and the model lacks training samples with high SWE, which eventually leads to the underestimation of high SWE by the machine learning model. Finally, in complex terrain, the integration of snow water equivalent products remains challenging.

### 3.2 Accuracy evaluation of the RRM SWE product at different altitudes

The accuracy of each snow water equivalent product is not absolute at different altitude gradients based on evaluations of the AMSR-E/AMSR2 SWE, ERA-Interim SWE, GLDAS SWE, GlobSnow SWE, and ERA5-land SWE products' accuracies (Fig. 5). The accuracy of a single snow water equivalent product is different from its overall accuracy. We consider the influence of altitude in the algorithm and make full use of the accuracy advantage of each snow water equivalent data for different altitude gradients.

The above verification results show that the MAEs between the RRM SWE dataset and measured snow water equivalent are 0.19, 0.23, 0.27, 0.26, 0.25, 0.21, 0.26, 0.21, 0.32, 0.31, and 0.21, the RMSEs are 6 mm, 26 mm, 32 mm, 30 mm, 30 mm, 16 mm, 11 mm, 7 mm, 33 mm, 32 mm, and 35 mm, the R values are 0.97, 0.87, 0.86, 0.81, 0.82, 0.96, 0.86, 0.81, 0.88, 0.79, and 0.81, and the $R^2$ values are 0.94, 0.75, 0.74, 0.65, 0.68, 0.91, 0.74, 0.65, 0.78, 0.62, and 0.66 at altitude gradients of <100 m, 100-200 m, 200-300 m, 300-400 m, 400-500 m, 500-600 m, 600-700 m, 700-800 m, 800-900 m, 900-1000 m and >1000 m, respectively (Fig. 5). Overall, RRM SWE product have the highest accuracy in the elevation interval <100 m, 100-200 m, 200-300 m, 400-500 m, 500-600 m, 600-700 m, 700-800 m, 800-900 m, and >1000 m. For the RRM SWE product itself, it has the best performance in the elevation interval <100 m. The ERA5-land product has the best performance in the elevation interval 300-400 m. The GlobSnow product has the best performance in the elevation interval 900-1000 m.

### 3.3 Comparison of spatial distribution patterns between the RRM SWE product and traditional snow water equivalent products

A comparison of the annual average snow water equivalent distributions is made between the RRM SWE and AMSR-E/AMSR2 SWE, ERA-Interim SWE, GLDAS SWE, GlobSnow SWE, and ERA5-land SWE in 2014, 2015, 2016, and 2017, and their spatial distribution patterns are shown in Fig. 6.

Overall, the RRM SWE dataset, AMSR-E/AMSR2 SWE dataset, ERA-Interim SWE dataset, GLDAS SWE dataset, GlobSnow SWE dataset, and ERA5-land SWE dataset have similar spatial distribution patterns in the Pan-Arctic region, showing a trend of lower snow water equivalent in low latitudes and higher snow water equivalent in high latitudes. The AMSR-E/AMSR2 SWE dataset covers a limited extent in the Pan-Arctic region, many data are missing, and low snow water equivalent values at low latitudes. In northern Siberia, the ERA-Interim SWE product has a higher snow water equivalent, and there are many abnormal, extreme snow water equivalent values (SWE > 500 mm) in this dataset. In low latitude regions, Alaska, North Siberia, and the easternmost region of Russia, the snow water equivalent of GLDAS SWE products is significantly lower. The GlobSnow SWE product lacks snow water equivalent data for Greenland, and this dataset has low snow water equivalents in the Baffin Island, the Koryak Mountains, the Kamchatka Peninsula, and Alaska regions. The ERA5-land SWE products have low snow water equivalents in northeastern Russia, Scandinavia, and northeastern Canada. The RRM SWE dataset is more reasonable for estimating the spatial distribution of snow water equivalent in the Pan-Arctic, and the data integrity is higher. Moreover, based on the new machine learning model, a variety of snow water equivalent data products in different time series are fused, which makes the RRM SWE dataset completely temporally and spatially continuous.

The relative difference between the RRM SWE data and GLDAS SWE data is the highest, and the relative difference is greater than 80% in most low altitude regions (Fig. 7). The relative difference between the RRM SWE data and the GlobSnow SWE data is relatively small overall, especially in most high latitude areas where the relative difference is less than 10% (Fig. 7). Overall, the annual average relative differences of the RRM SWE data and AMSR2 SWE, ERA-Interim SWE, GLDAS SWE, GlobSnow SWE, and ERA5-land SWE are 39%, 41%, 49%, 26%, and 33%, respectively (Fig. 7). Previous studies have shown that the accuracy of snow water equivalent in the Northern Hemisphere estimated by GlobSnow SWE data is higher (Pulliainen et al., 2020), while the spatial distribution pattern of the RRM SWE data is close to the estimation result of

GlobSnow SWE. In addition, the single point verification results based on the measured snow water equivalent data of meteorological stations in section 4.1 show that the RRM SWE dataset has higher accuracy than the GlobSnow SWE dataset. The RRM SWE dataset has good accuracy.

### 3.4 Comparison of the annual variation tendencies of AMSR-E/AMSR2 SWE, ERA-Interim SWE, GLDAS SWE, GlobSnow SWE, and ERA5-land SWE and the RRM SWE in the Pan-Arctic region

Based on the Mann-Kendall trend test, we analyzed the changing trend in the annual average snow water equivalent of the AMSR-E/AMSR2 SWE, ERA-Interim SWE, GLDAS SWE, GlobSnow SWE, ERA5-land SWE, and RRM SWE in the Pan-Arctic region from 1979 to 2019.

Based on the Mann-Kendall trend test (see Fig. 8 and Table 3), from 1979 to 2019, the test value of the ERA-Interim annual average snow water equivalent is 1.08, and there is no significant change trend under the significance test level of 0.05. The test value of the GLDAS annual average snow water equivalent was 4.95 and showed a significant increasing trend at the significance test level of 0.05. The test values of the AMSR-E/AMSR2 annual average SWE, GlobSnow annual average SWE, ERA5-land annual average SWE, and RRM annual average SWE are -3.26, -2.54, -3.43, and -2.95, respectively, and these four SWEs showed a significantly decreasing trend at the significance test level of 0.05. Based on the analysis of the RRM SWE product, between 1979 and 2019, the annual average snow water equivalent in the Pan-Arctic decreased by 15.1 percent. In the Northern Hemisphere, spring snow cover extent has decreased significantly, according to the Fifth Assessment Report (AR5) of the IPCC. Between 1967 and 2010, the spring snow cover extent decreased by an average of 1.6 percent per decade, while the June snow cover extent decreased by 11.7 percent per decade (Stocker, 2014). Most studies have shown that the annual variation tendency of snow depth and snow cover extent showed a significant decreasing trend in the Northern Hemisphere (Brutel-Vuilmet et al., 2013), which is consistent with the annual variation tendency of the RRM SWE dataset. This dataset can reflect the characteristics of snow cover change in the Pan-Arctic under the background of climate change and can be used as the driving data for the climate model to support climate change-related research. In addition, this dataset is expected to provide a snow data basis for the study of "Arctic amplification".

## 4 Data availability

The RRM SWE product is available for free download from the 'A Big Earth Data Platform for Three Poles' (http://dx.doi.org/10.11888/Snow.tpdc.271556) (Li et al., 2021). The temporal resolution of RRM SWE product is daily, and the spatial resolution is 10 km. It spans latitude 45°N-90°N and longitude 180°W-180°E. A brief summary and data description document (includes data details, spatial range, usage method and etc.) are also provided.

## 5 Conclusions

In this study, we propose a method to fuse multisource snow water equivalent data by a ridge regression model based on machine learning. A new method was utilized to prepare a set of spatiotemporal seamless snow water equivalent datasets of RRM SWE, combined with the original AMSR-E/AMSR2 SWE dataset, ERA-Interim SWE dataset, GLDAS SWE dataset, GlobSnow SWE dataset, and ERA5-land SWE dataset. In the RRM SWE dataset, the time series of the data is 1979-2019, the temporal resolution is daily, the spatial resolution is 10 km, and the spatial range is the Pan-Arctic region.

The RRM SWE data product has the best accuracy, especially for the estimation of low snow water equivalent. The accuracy ranking of the snow water equivalent dataset verified by the test dataset is described as follows: RRM SWE > GlobSnow SWE > ERA5-land SWE > AMSR-E/AMSR2 SWE > ERA-Interim SWE > GLDAS SWE. The accuracy of the RRM SWE dataset is higher than that of the existing snow water equivalent products at most elevation intervals. Moreover, the RRM SWE dataset fills in the missing data of the original snow water equivalent dataset spatiotemporally.

Compared with traditional fusion methods, machine learning methods have a good advantage. We find that the simple machine learning algorithm not only has high efficiency but also has good accuracy in the preparation of snow water equivalent products on a global scale. Without losing the advantages of existing snow water equivalent products, this method can also make full use of station observation data to integrate the advantages of various snow water equivalent products. The model training process does not rely too much on a specific sample, and this model has a strong generalization ability. In addition, the influence of altitude on the preparation scheme is considered in detail in the model. Compared with the snow water equivalent dataset prepared by the traditional method, the spatial resolution is only 25 km, while this new method obtains a

snow water equivalent dataset with a higher spatial resolution of 10 km.

We propose that the RRM SWE dataset preparation scheme has good continuity and can prepare real-time and high-quality snow water equivalent datasets in the Pan-Arctic. In addition, the new method proposed in this paper has the advantages of simplicity and high precision in preparing large-scale snow water equivalent datasets and can be easily extended to the preparation of other snow datasets. This dataset is an important supplement to the Pan-Arctic snow water equivalent database and is expected to provide data support for Arctic cryosphere studies and global climate change studies.

**Author contribution.**

DS and HL designed the study and wrote the manuscript; JW, XH, and TC contributed to the discussions, edits, and revisions. DS and WJ compiled the model code.

**Competing interests.**

The authors declare that they have no conflict of interest.

**Acknowledgements.**

The authors would like to thank the European Space Agency (ESA) for GlobSnow data, the European Centre for Medium-Range Weather Forecasts (ECMWF) for ERA-Interim data and ERA5-land data, the National Aeronautics and Space Administration (NASA) for AMSR-E/AMSR2 data, the Goddard Earth Sciences Data and Information Services Center (GES DISC) for GLDAS data, the Russian Federal Service For Hydrometeorology and Environmental Monitoring (ROSHYDROMET) for snow survey data, and the National Oceanic and Atmospheric Administration (NOAA) for GHCN-Daily SWE data.

**Financial support.**

This research was supported by the Strategic Priority Research Program of the Chinese Academy of Sciences (Grant No. XDA19070302), the National Science Fund for Distinguished Young Scholars (Grant No. 42125604), the National Natural Science Foundation of China (Grant No. 41971399, 41971325, 42171391).

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

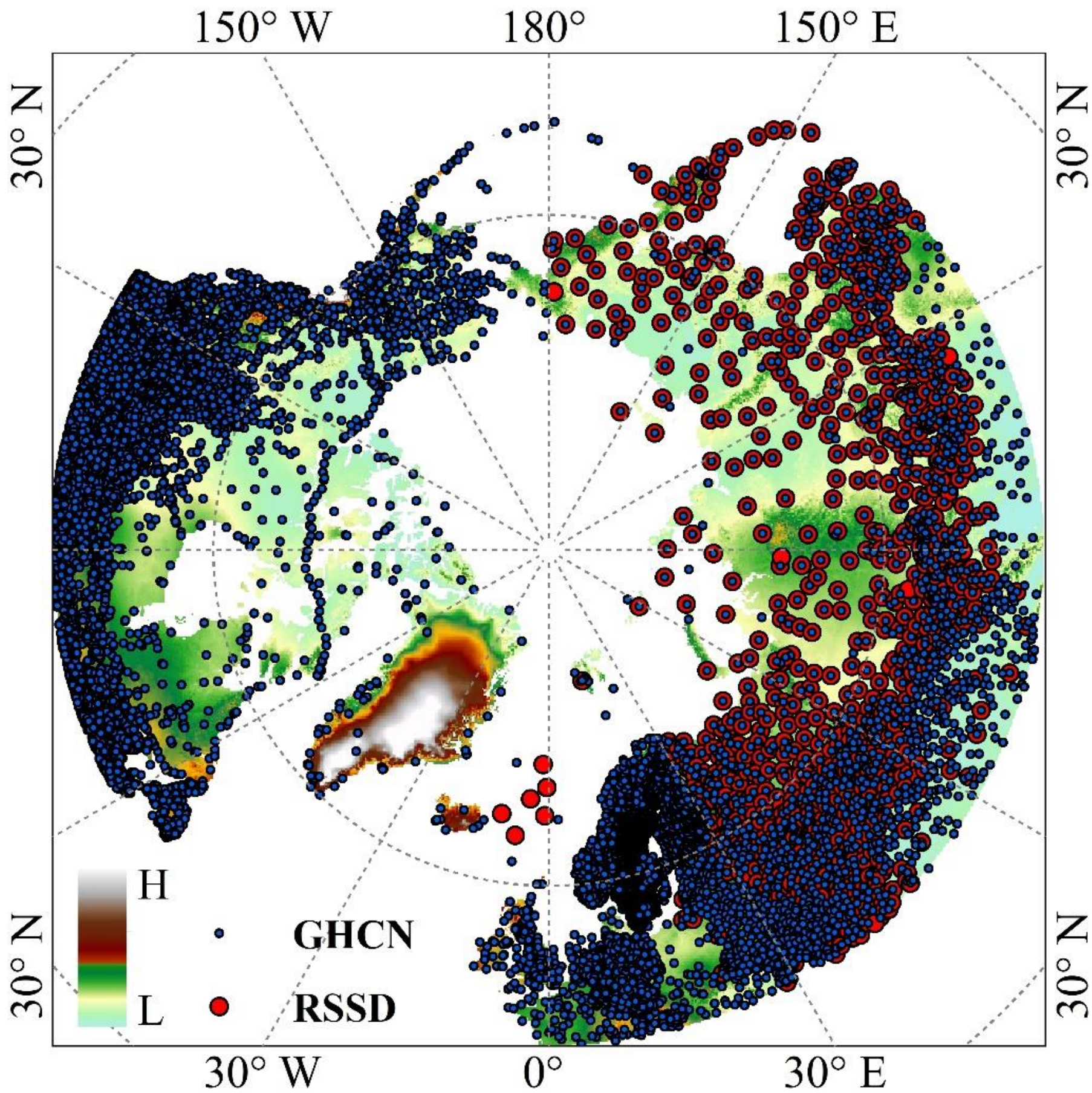

**Figure 1: The DEM and snow survey stations of the research region (GHCN is the Global Historical Climatology Network station, and RSSD is the Russian snow survey station).**

**Table 1: Introduction to the SWE data.**

| Data type | Data name | Time series | Temporal resolution | Spatial resolution | Spatial coverage | File format |
|---|---|---|---|---|---|---|
| Remote sensing data | AMSR-E/AMSR2 | 2002-2011/2012-2020 | Daily | 25 km x 25 km | Global (No Greenland) | HDF5 |
| Data assimilation dataset | GLDAS | 1979-2020 | Daily | 0.25°×0.25° | Global | NetCDF |
| Reanalysis dataset | GlobSnow | 1979-2018 | Daily | 0.25°×0.25° | Northern Hemisphere (No Greenland) | NetCDF |
| | ERA-Interim | 1979-2019 | Daily | 0.25°×0.25° | Global | NetCDF |
| | ERA5-land | 1981- present | Hour | 0.1°×0.1° | Global | NetCDF |

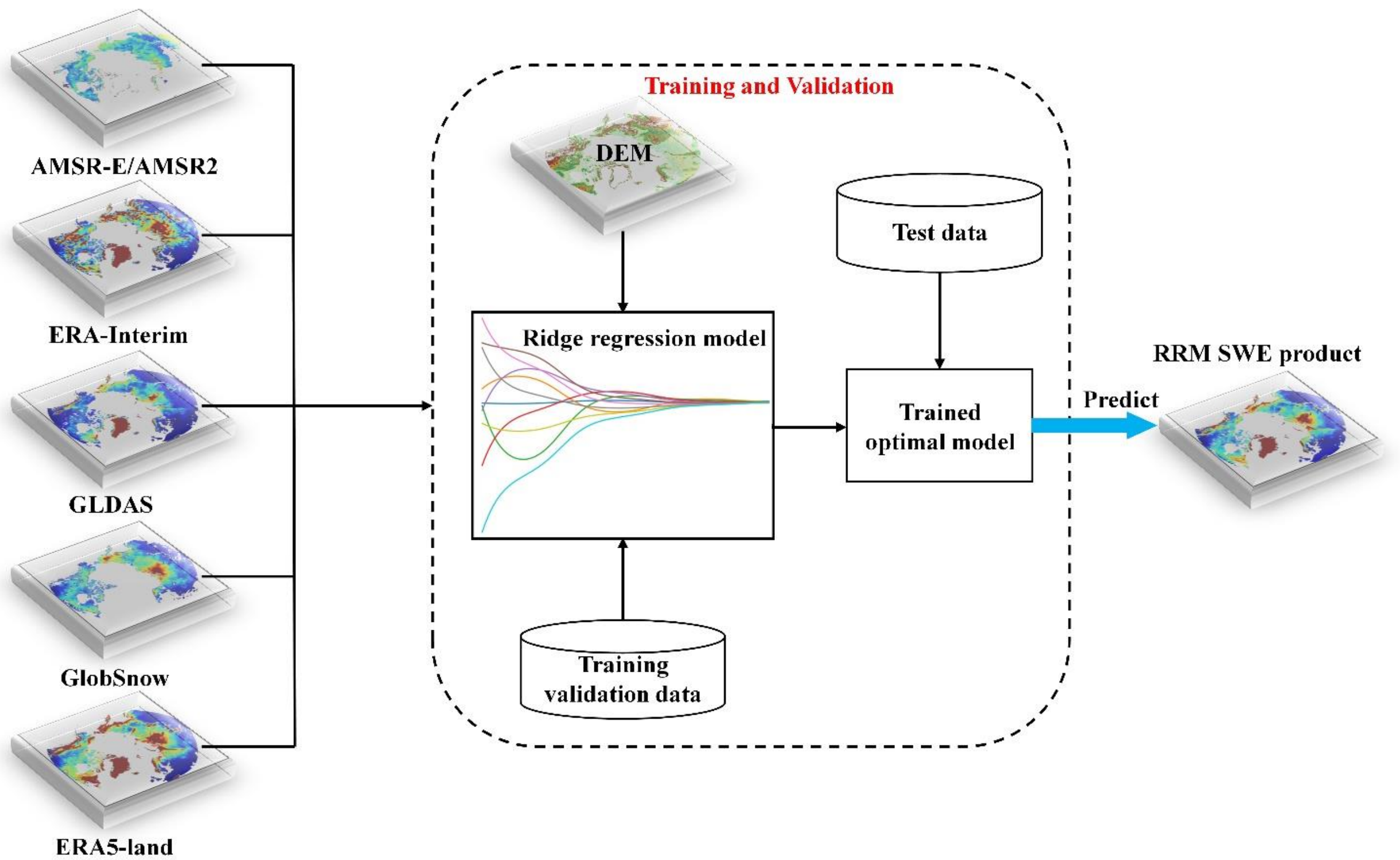

**Figure 2: Flow chart of RRM SWE data preparation (preparation of spatiotemporal seamless snow water equivalent datasets mainly includes three processes: model training, model reasoning, and snow water equivalent data preparation).**

**Table 2: Error list for the station data and grid snow water equivalent products.**

| Error type | MAE | RMSE (mm) | R | $R^2$ |
|---|---|---|---|---|
| ERA-Interim | 0.49 | 51.98 | 0.62 | 0.39 |
| AMSR-E/AMSR2 | 0.48 | 47.51 | 0.56 | 0.31 |
| GLDAS | 0.56 | 62.84 | 0.41 | 0.17 |
| GlobSnow | 0.31 | 35.21 | 0.75 | 0.56 |
| ERA5-land | 0.34 | 38.11 | 0.84 | 0.70 |
| RRM SWE | 0.24 | 30.29 | 0.87 | 0.76 |

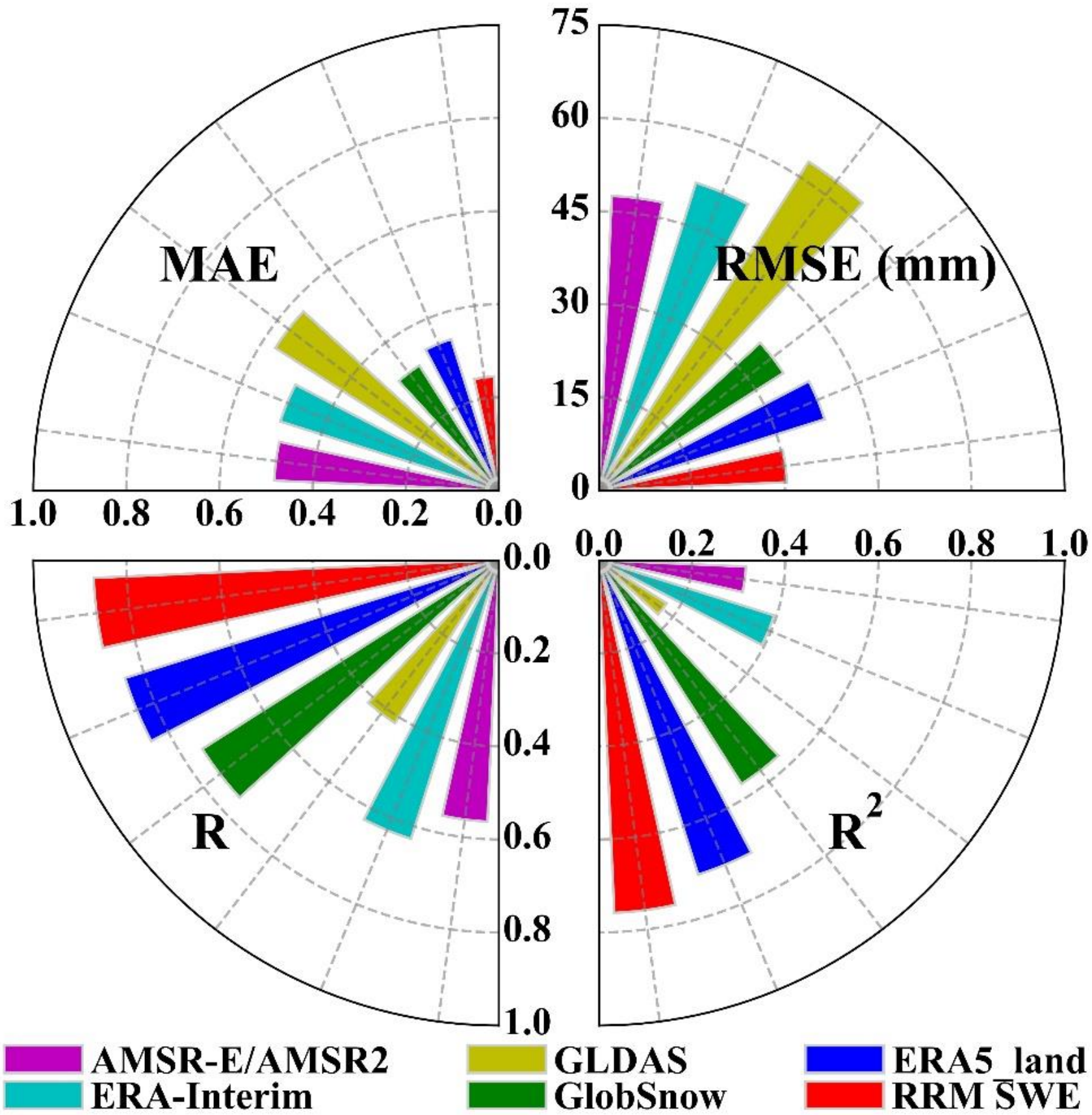

**Figure 3: Accuracy comparison of various snow water equivalent products. The upper left sector represents MAE, the upper right sector represents RMSE, the lower-left sector represents R, and the lower right sector represents $R^2$. The sector axis represents the size of the error, and the color represents different snow water equivalent datasets.**

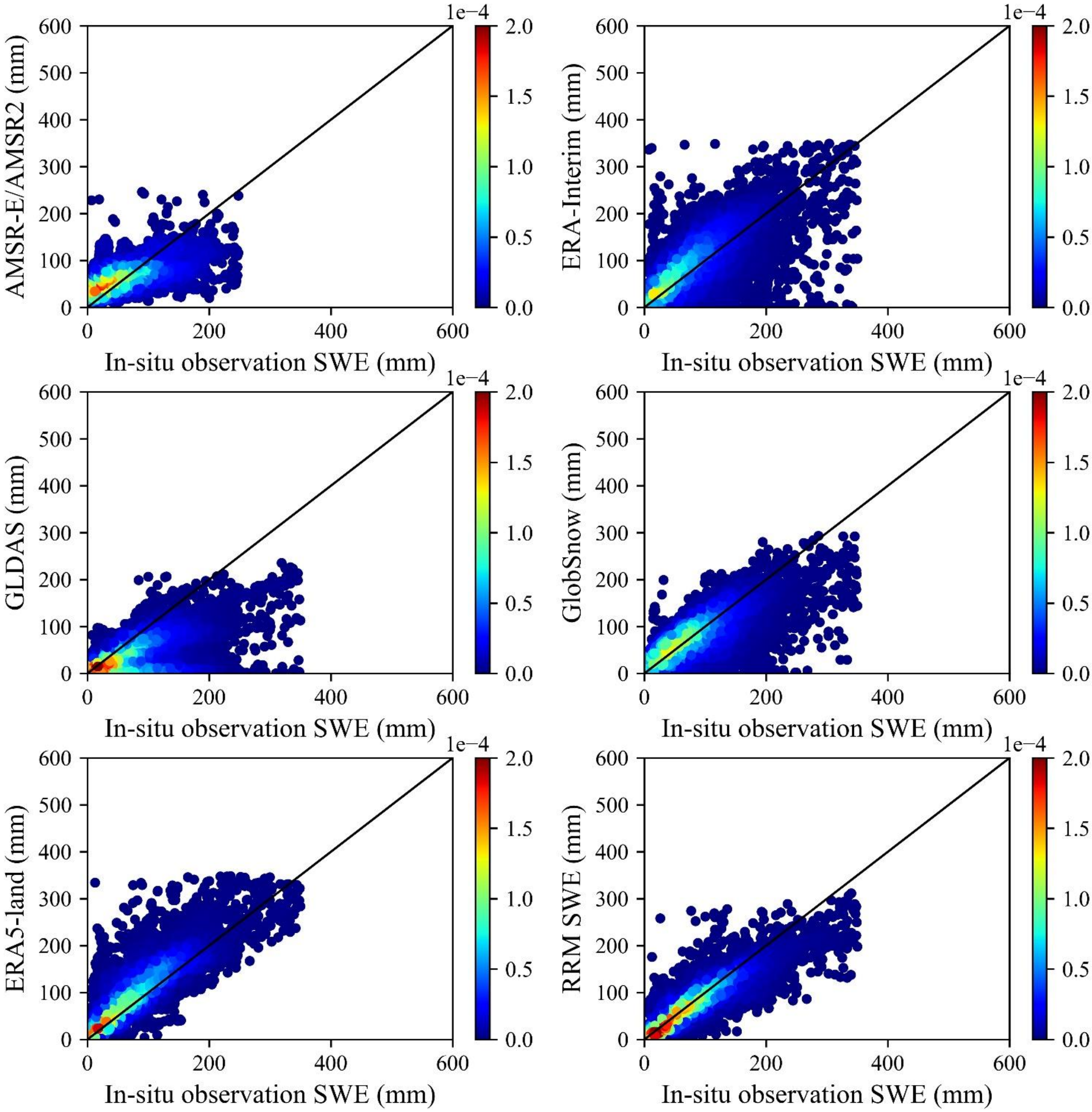

**Figure 4: Error verification density diagram (A total of 8618 sample points were used for verification.). The color bar represents the value of kernel density estimation. The closer the high-density area is to the 1:1 line, the higher the verification accuracy of the dataset is at most of the measuring stations.**

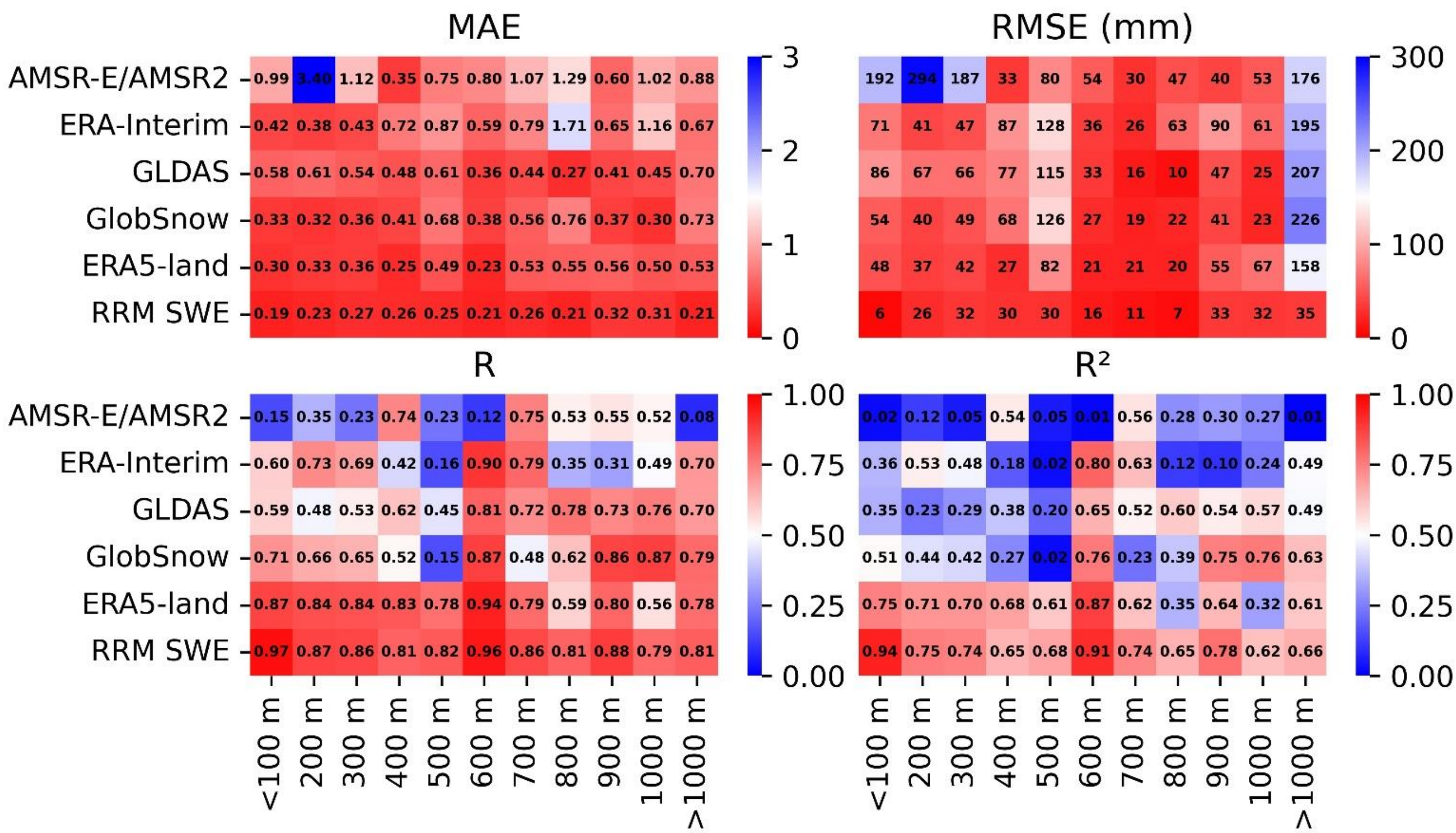

**Figure 5: Comparison of the error between the RRM SWE and AMSR-E/AMSR2 SWE, ERA-Interim SWE, GLDAS SWE, GlobSnow SWE, and ERA5-land SWE at different altitudes (the abscissa represents the altitude gradient, and the ordinate represents different snow water equivalent datasets). The color bar indicates the error in each snow water equivalent dataset. The closer to red the color is, the higher the accuracy is. MAE: mean absolute error, RMSE: root mean square error, R: Pearson's correlation coefficient, $R^2$: coefficient of determination).**

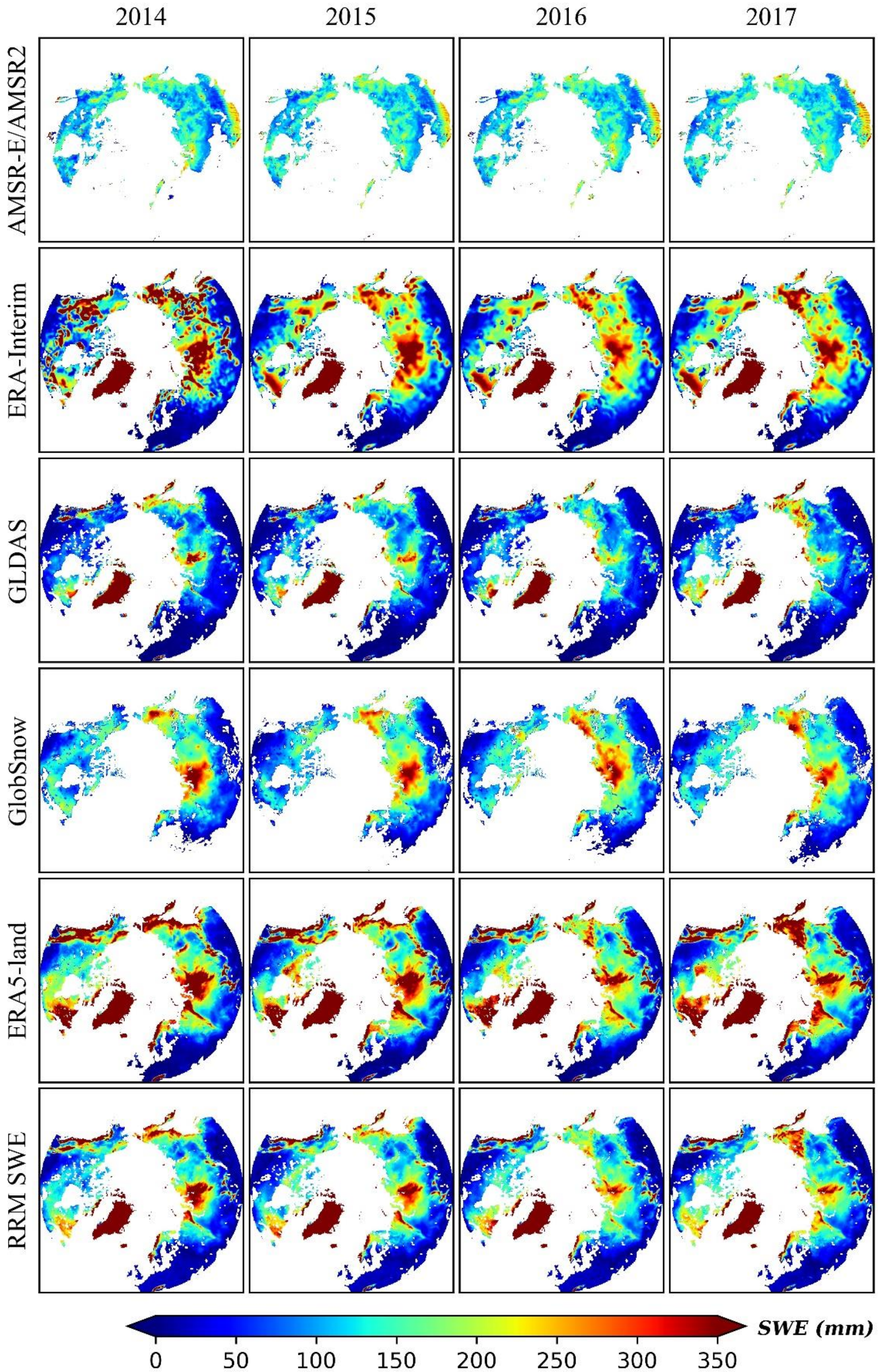

497

**Figure 6: Comparison of the spatial distribution characteristics between the RRM SWE and AMSR-E/AMSR2 SWE, ERA-Interim SWE, GLDAS SWE, GlobSnow SWE, and ERA5-land SWE (the four columns of images represent the comparison results in 2014, 2015, 2016, and 2017, respectively).**

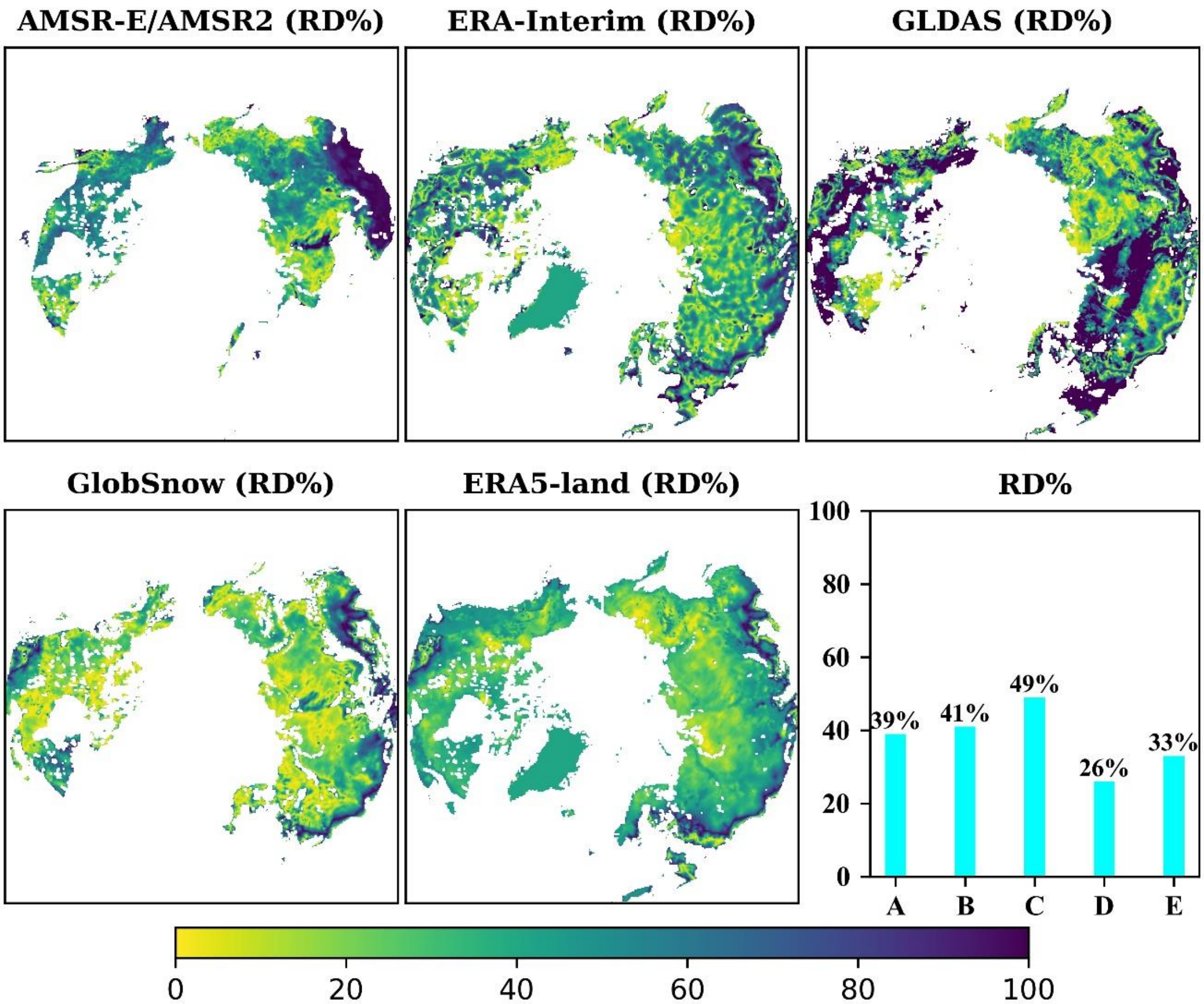

**Figure 7: Temporal and spatial distributions of relative differences (RD%) between the RRM SWE and AMSR-E/AMSR2 SWE, ERA-Interim SWE, GLDAS SWE, GlobSnow SWE, and ERA5-land SWE. Lower right subgraph: Comparison of annual average relative differences between the RRM SWE and AMSR2 SWE (A), ERA-Interim SWE (B), GLDAS SWE (C), GlobSnow SWE (D), and ERA5-land SWE (E).**

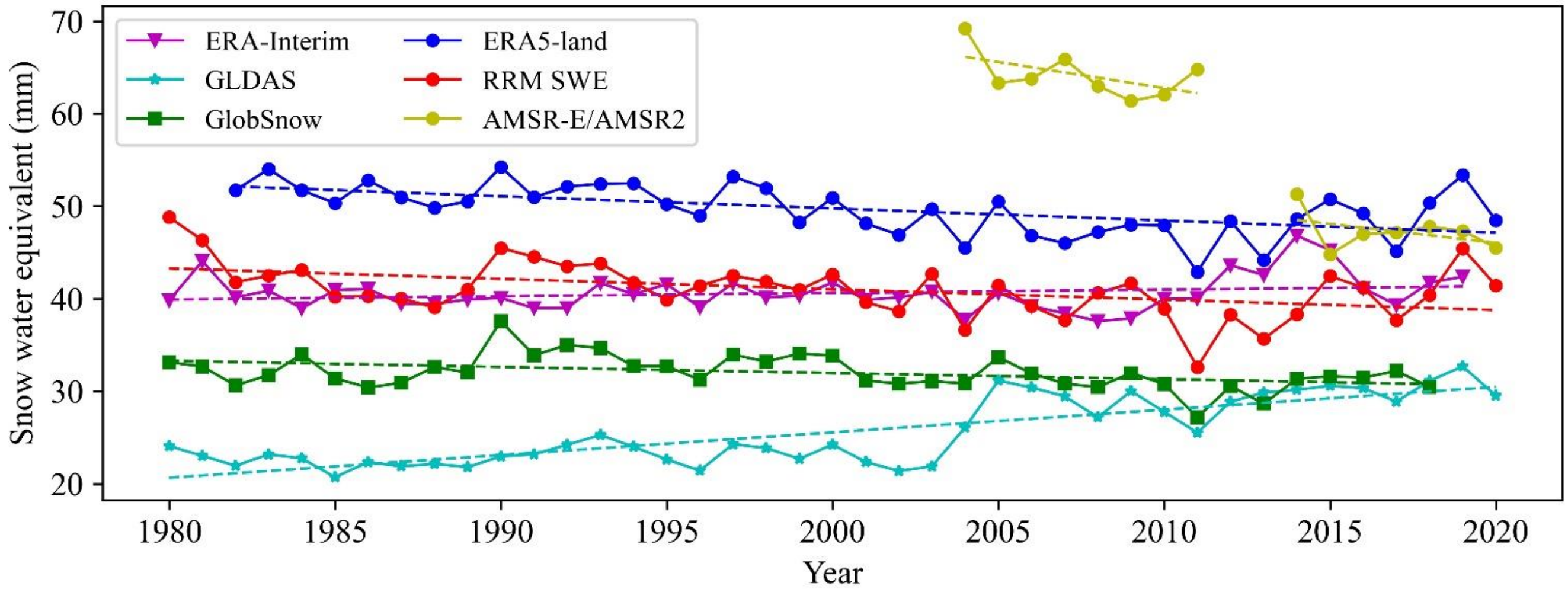

**Figure 8: Annual variation tendency in the AMSR-E/AMSR2 SWE, ERA-Interim SWE, GLDAS SWE, GlobSnow SWE, and ERA5-land SWE and RRM SWE products from 1979 to 2019 (the dotted line is the trend line calculated based on the Mann-Kendall method).**

512 **Table 3: Results of the Mann-Kendall trend test performed for various snow water equivalent products for 1979 to 2019.**

| Data | P-value | Test value | Trend |
|---|---|---|---|
| AMSR-E/AMSR2 | 0.00 | -3.26 | Decreasing |
| ERA-Interim | 0.27 | 1.08 | No trend |
| GLDAS | 7.29e-07 | 4.95 | Increasing |
| GlobSnow | 0.01 | -2.54 | Decreasing |
| ERA5-land | 0.00 | -3.43 | Decreasing |
| RRM SWE | 0.00 | -2.95 | Decreasing |

513 *Significance level alpha = 0.05