# Peer review of "Reconstruction of a daily gridded snow water equivalent product for the land region above 45° N based on a ridge regression machine learning approach"

_Earth System Science Data, 2021_

## Author Comment (AC1)

**Response to Anonymous Referee #1**

We greatly appreciate your valuable suggestions and comments. We studied your comments carefully and made substantial revisions accordingly. We hope that our corrections meet the requirements. Our modifications and responses to the reviewers' comments are listed as follows. The reviewers' comments are in blue text, while the authors' responses are in black text. All revisions in the revised manuscript are in red text.

The idea of the paper is to combine various data records on snow water equivalent (SWE) with ridge regression model (RRM) technique. This is an interesting and welcomed approach that possibly provides improved estinates on SWE of the northern hemisphere. However there are some issues that need further consideration and clarification.

**Reply: We greatly appreciate your careful revision of our manuscript.**

1.Apperently, the paper applies point-wise observations on snow cover, not distributed observations from snow courses. Thus the reference data does not directly describe the SWE on the scale of data products. Moreover, the NOAA GHCN data does not include direct measurements on SWE (they are evidently based on snow depth observations at meteorological stations). Thus, the applied reference data may include systematic errors due to inadeqate consideration of temporally and spatially varying snow bulk density.

**Reply: Thank you for your very helpful suggestion.**

We fully agree with your opinion that the NOAA GHCN SWE data do have some systematic errors. Therefore, we discarded the NOAA GHCN SWE data. In addition, the Russian snow survey data (RSSD) are also a snow course data type. Therefore, we retrained, tested, and validated the model using new hemispheric-scale snow course observational (HSSC) data and RSSD to generate a new version of the RRM SWE product and re-evaluated the product.

**Most of the contents in the article have been revised due to the replacement of training data, and the detailed changes have been marked in red in the revised manuscript.**

**Abstract:**

[revised manuscript text omitted]
 authors should also note a limitation that reanalysis data-based products (that are used here together with obsrvational data-based products) on SWE do not provide model-independent data (results here are not independent on models).

**Reply: The text has been modified to improve clarity.**
Yes, we fully agree with your comments. The reanalyzed data based on the gridded SWE product do suffer from the problem that the data and the model are not independent. The reason for this is that the model estimation is distorted or difficult to estimate accurately due to the existence of an exact correlation or high correlation between the SWE target variable and SWE predictor variables in the training model, namely, the so-called multicollinearity problem. The ridge regression model can solve this multicollinearity problem well, i.e., the independence of models and SWE products based on observational data. The reason why the ridge regression model is selected for RRM SWE product preparation in this study is to solve this problem.

**We have provided detailed supplementary explanations in Section 2.3.**
"The ridge regression model is flexible in the choice of predictor variables and does not require the predictor and target variable to be independent of each other. It can effectively solve the multicollinearity problem of predictor and target variables as well as reduce the impact of this problem on the training model (Duzan and Shariff, 2015; Saleh et al., 2019). Generally, since the reanalysis data based on SWE products cannot make the products and models independent of each other, i.e., they are prone to the multicollinearity problem, which leads to distorted model estimation or difficulty in performing accurate estimations. In contrast, the ridge regression model can successfully solve the multicollinearity problem, i.e., the independence of training products and models. In addition, when integrating multiple SWE products, the accuracy of each SWE dataset is likely to differ. A small change in one of the SWE products involved in the training will cause a significant error in the final calculation results, while the ridge regression model has high accuracy and stability for this "ill-conditioned" SWE data".

**References**
Duzan, H. and Shariff, N. S. B. M.: Ridge regression for solving the multicollinearity problem: a review of methods and models, Journal of Applied Science, 2015.
Saleh, A. M. E., Arashi, M., and Kibria, B. G.: Theory of ridge regression estimation with applications, John Wiley & Sons2019.

Detailed comments:
3. Title: Since the study region includes all land areas north of the latitude of 45 degrees, the term

Pan-Arctic is not good.

**Reply: This suggestion has been accepted, and the text has been modified.**

We have modified 'Pan-Arctic' as 'land region above 45° N'.

4.Lines 59-61 and 96-98: The authors should note that a recent investigation (Pulliainen et al. 2020) applied a bias correction to GlobSnow and reanalysis data products to obtain improved estimates on peak annual regional and hemispehric snow mass/SWE. This was done using hemispheric snow course obervations.

**Reply: The text has been modified to improve clarity.**

We have noted this very interesting study and used the latest version of the GlobSnow SWE data. In addition, we have added a description of this study to the Introduction and Data sections.

"An effective method was applied in a study by Pulliainen et al (Pulliainen et al., 2020), who applied a bias correction to GlobSnow and reanalysis data products based on SWE snow course measurements to obtain improved estimates on annual peak snow mass and SWE in the Northern Hemisphere."

Pulliainen, J., Luojus, K., Derksen, C., Mudryk, L., Lemmetyinen, J., Salminen, M., Ikonen, J., Takala, M., Cohen, J.,Smolander, T., and Norberg, J.: Patterns and trends of Northern Hemisphere snow mass from 1980 to 2018 (vol 41, pg 861,2020), Nature, 10.1038/s41586-020-2416-4, 2020.

5.Lines 139-166: The method should be explained in more detail. In Eq. (1) p probably includes as variables the applied five SWE products, but how is the DEM considered? What is the number of samples N in the training data set? How is the training and testing data set defined/selected?

**Reply: Thank you for your good suggestion. We have added details to the methods section.**

In this study, the DEM was used as an important environmental feature information input to the ridge regression model and was utilized in the model training. The DEM is an auxiliary terrain feature variable in addition to the five SWE prediction feature variables, namely, AMSR-E/AMSR2 SWE, ERA-Interim SWE, GLDAS SWE, GlobSnow SWE, and ERA5-land SWE.

The total number of samples N in the training dataset was 271651. The sample sizes of the training dataset, validation dataset and test dataset were divided according to the ratio of 7:2:1, where the training, validation, and test set sample numbers were 271651, 77614 and 38807, respectively.

**We have made detailed modifications in Section 2.3.**

"DEM was used as an important environmental feature input to the ridge regression model and was included in the model training. DEM is an auxiliary terrain feature variable in addition to the five SWE prediction feature variables, AMSR-E/AMSR2 SWE, ERA-Interim SWE, GLDAS SWE, GlobSnow SWE, and ERA5-land SWE."

$$\hat{\beta}^{ridge} = \operatorname*{argmin}_{\beta} \left\{ \sum_{i=1}^{N} \left( y_i - \beta_0 - \sum_{j=1}^{p} x_{ij}\beta_j \right)^2 + \lambda \sum_{j=1}^{p} \beta_j^2 \right\}, \tag{1}$$

where $\hat{\beta}^{ridge}$ is the extremum solution function of ridge regression and $p$ is the number of gridded

SWE product variables involved in training. $x_i$ are the prediction feature variables, which contain two

parts, one set contains the main feature variables of the gridded SWE products, and the other part consists of the DEM auxiliary feature variables. $y_i$ is the observed SWE, and $\lambda$, $\beta$, $\beta_j$ and $\beta_0$ are the parameters to be solved. $1, \cdots, N$ is the sample of the training dataset. $\lambda \sum_{j=1}^{p} \beta_j^2$ is the penalty function terms. The total number of samples $N$ in the training dataset is 271651. The sample sizes of the training data set, validation data set and test data set are divided according to the ratio of 7:2:1, where the numbers of training set, validation set and test set samples are 271651, 77614 and 38807, respectively."

6.Lines 278-279 and 306-308: How is the annual average SWE calculatated, mean value across the whole year?

**Reply: This suggestion has been accepted, and the text has been modified accordingly.**

The "annual average SWE" in Lines 278-279 represents the spatially annual average SWE over the land region above 45° N.

The "annual average SWE" of Lines 306-308 represents a further averaging of the annual average SWE across the land region above 45° N over all pixels. Here, the "annual average SWE" $\overline{SWE}$ can be expressed by the following formula:

$$\overline{SWE} = \frac{\sum_{i}^{N} S_i \Big/ N}{n}$$

where $S_i$ is the spatially annual average SWE over the land region above 45° N, $N$ is the year, and $n$ is all the image elements in the land region above 45° N.

To avoid confusion, we changed the "annual average SWE" on Lines 278-279 to "spatially distributed annual average SWE". In Lines 306-308, "annual average SWE" was modified to "region-wide annual average SWE".

7.Lines 467-468 (Fig. 1): Some data stations (RSSD) are located in the open Norwegian Sea, and transect of stations in Canada is odd. Are these observations from some campaign?

**Reply: This suggestion has been accepted, and Figure 1 has been modified accordingly.**

We double-checked the station locations in the RSSD dataset and found that observational sites do exist on some of the islands in the Norwegian Sea, so the locations of the stations in Figure 1 are reasonable. According to your suggestion, we have switched to using "hemispheric-scale snow course observational data", so the issue of the "transect of stations" in the Canadian region has been resolved.

[Figure]

Figure 1: The DEM and snow survey stations of the research region. The right subgraph shows the DEM and the left subgraph shows the SWE observational stations. HSSC, hemispheric-scale snow course; RSSD, the Russian snow survey station. The spatial range of the RRM SWE product is consistent with that of DEM).

8.Lines 508-510: The wide spread of range is y-axis is odd.

**Reply: Thank you for your suggestion, and Figure 8 has been modified accordingly.**

[Figure]

Figure 8: Annual variation tendency in the AMSR-E/AMSR2 SWE, ERA-Interim SWE, GLDAS SWE, GlobSnow SWE, ERA5-land SWE and RRM SWE products from 1979 to 2019 (the dotted line is the trend line calculated based on the Mann-Kendall method).

9.References:

Pulliainen, J,, Luojus, K., Derksen, C., et al. Patterns and trends of Northern Hemisphere snow mass from 1980 to 2018, Nature, 581: 294-298, 2020.

**Reply: Thank you for providing this reference, which we have cited in the manuscript. We studied the reference carefully and discussed it in detail in the manuscript.**

---

## Author Comment (AC2)

**Response to Anonymous Referee #2**

We greatly appreciate your valuable suggestions and comments. We studied your comments carefully and made substantial revisions accordingly. We hope that our revisions meet the requirements. Our modifications and responses to the reviewer's comments are listed as follows. The reviewer's comments are in blue text, while the authors' responses are in black text. All revisions in the revised manuscript are in red text.

A very interesting paper to develop an improved SWE dataset based on existing multi-source SWE products using machine learning. However, there are some points which need to be well addressed. Please see following comments:

**Reply: We greatly appreciate your careful review of our manuscript.**

1. The ridge regression machine learning algorithm in Section 2.3 needs to be better described.

■ Why ridge regression is selected among so many machine learning algorithms? The advantages of using ridge regression over others for this application need to be highlighted.

**Reply: The text has been modified to improve clarity.**

The advantages of the ridge regression model are the following:

1) The ridge regression algorithm can effectively solve the multicollinearity problem, i.e., the independence of training products and models. In this study, the reanalysis data based on SWE products cannot make the products and models independent of each other, i.e., they are prone to the multicollinearity problem, which leads to distorted model estimation or difficulty in accurate estimation. It is, therefore, a good way to use the ridge regression algorithm to address this problem.

2) The ridge regression model has high accuracy and stability for "ill-conditioned" data. In this study, we used various forms of gridded SWE datasets, and the accuracy of each gridded SWE dataset varied. A small change in one of the SWE products involved in the training will cause a large error in the final calculation results. Therefore, using the ridge regression algorithm to address "ill-conditioned" SWE data is an effective method.

**In addition, we provide an additional explanation of the advantages of the ridge regression model in Section 2.3.**

"Generally, since the reanalysis data based on SWE products cannot make the products and models independent of each other, i.e., they are prone to the multicollinearity problem, which leads to distorted model estimation or difficulty in performing accurate estimations. In contrast, the ridge regression model can successfully solve the multicollinearity problem, i.e., the independence of training products and models. In addition, when integrating multiple SWE products, the accuracy of each SWE dataset is likely to differ. A small change in one of the SWE products involved in the training will cause a significant error in the final calculation results, while the ridge regression model has high accuracy and stability for this "ill-conditioned" SWE data".

■ What is the requirement of ridge regression in terms of variable dependence? Does it require predictor variables and target variable independent? Are the station data from GHCN or Russian snow survey used (or partially used) for generating the gridded SWE products used in this paper? If so, the target variable and predictor variables are intrinsically related. How does this impact this

algorithm? Please provide more information to explain and justify.

**Reply: The text has been modified to improve clarity.**

**Regarding the question "**What is the requirement of ridge regression in terms of variable dependence? Does it require predictor variables and target variable independent?**, the ridge regression model is flexible in the choice of predictor variables and does not require the predictor and target variable to be independent of each other. It is effective in solving machine learning problems with correlations between predictor and target variables (Duzan and Shariff, 2015; Saleh et al., 2019).

**Regarding the question "**Are the station data from GHCN or Russian snow survey used (or partially used) for generating the gridded SWE products used in this paper?**", a few of the gridded SWE products used in this paper use hemispheric-scale snow course (HSSC) observations (in this revised version, the GHCN data have been replaced by HSSC data) data and Russian snow survey data (RSSD), and the predictor and target variables are intrinsically related.

**Regarding the question "**How does this impact this algorithm?**", this "intrinsically related" does reduce the accuracy of SWE estimates and leads to a reduction in the stability of SWE estimates. However, most of the current SWE observations are inevitably subject to such a multicollinearity problem. The ridge regression model is a biased estimation method specifically designed for the problem of multicollinearity data (Duzan and Shariff, 2015; Saleh et al., 2019). It has good tolerance to "ill-conditioned" data (Hoerl and Kennard, 1970b; Guilkey and Murphy, 1975). The ridge regression model can solve the multicollinearity problem of predictor and target variables and effectively reduce the impact of this problem on the preparation accuracy of the RRM SWE product.

**We have provided detailed supplementary explanations in the Introduction and Section 2.3.**
**Introduction:**
"The ridge regression model is a biased estimation method specifically designed to address the problem of multicollinear data (Duzan and Shariff, 2015; Saleh et al., 2019). It has good tolerance to "ill-conditioned" data and has a good effect in using SWE data to address the multicollinearity problem (Hoerl and Kennard, 1970b; Guilkey and Murphy, 1975)."

**Section 2.3.:**
"The ridge regression model is flexible in the choice of predictor variables and does not require the predictor and target variable to be independent of each other. It can effectively solve the multicollinearity problem of predictor and target variables as well as reduce the impact of this problem on the training model (Duzan and Shariff, 2015; Saleh et al., 2019). Generally, since the reanalysis data based on SWE products cannot make the products and models independent of each other, i.e., they are prone to the multicollinearity problem, which leads to distorted model estimation or difficulty in performing accurate estimations. In contrast, the ridge regression model can successfully solve the multicollinearity problem, i.e., the independence of training products and models. In addition, when integrating multiple SWE products, the accuracy of each SWE dataset is likely to differ. A small change in one of the SWE products involved in the training will cause a significant error in the final calculation results, while the ridge regression model has high accuracy and stability for this "ill-conditioned" SWE data".

**References**

Hoerl, A. E. and Kennard, R. W.: Ridge regression: Biased estimation for nonorthogonal problems, Technometrics, 12, 55-67,1970

Guilkey, David K., and James L. Murphy.: Directed ridge regression techniques in cases of multicollinearity, Journal of the American Statistical Association, 70, 769-775, 1975

Duzan, H. and Shariff, N. S. B. M.: Ridge regression for solving the multicollinearity problem: review of methods and models, Journal of Applied Science, 2015.

Saleh, A. M. E., Arashi, M., and Kibria, B. G.: Theory of ridge regression estimation with applications, John Wiley & Sons2019.

■ It is unclear how DEM is used in the ridge regression. Is it used as a predictor variable in addition to AMSR-E/AMSR2, ERA-Interim, GLDAS, GlobSnow, ERA5-land SWE or just used to evaluate different model performances related to elevation? There are many important factors impacting the SWE estimation in addition to DEM. Have you also considered adding more variables to further improve the model? Please provide more information to explain.

**Reply: The text has been modified to improve clarity.**

In this study, the DEM was used as an important environmental feature information input to the ridge regression model and was included in the model training. The DEM is an auxiliary terrain feature variable in addition to the five SWE prediction feature variables AMSR-E/AMSR2 SWE, ERA-Interim SWE, GLDAS SWE, GlobSnow SWE, and ERA5-land SWE.

In addition to the DEM, meteorological elements, NDVI, land type, and other factors will affect the SWE estimation. Unfortunately, our current training model does not consider these factors in detail, which is a limitation of the current RRM SWE product. Currently, there are two main reasons why we do not consider these impact factors. The uncertainty of the model will be greater after adding these impact factors, and the accuracy of the RRM SWE product may not be improved. However, an investigation considering these impact factors in detail is a significant workload and cannot currently be performed. In future studies, we will carefully consider these factors in detail to further improve the accuracy of the RRM SWE product.

**We added a description of the role of DEM data in Section 2.3.**

"The digital elevation model (DEM) was used as an important environmental feature input to the ridge regression model and was included in the model training. The DEM is an auxiliary terrain feature variable in addition to the five SWE prediction feature variables, AMSR-E/AMSR2 SWE, ERA-Interim SWE, GLDAS SWE, GlobSnow SWE, and ERA5-land SWE."

$$\hat{\beta}^{ridge} = \underset{\beta}{\arg\min} \left\{ \sum_{i=1}^{N} \left( y_i - \beta_0 - \sum_{j=1}^{p} x_{ij}\beta_j \right)^2 + \lambda \sum_{j=1}^{p} \beta_j^2 \right\}, \tag{1}$$

where $\hat{\beta}^{ridge}$ is the extremum solution function of ridge regression and $p$ is the number of gridded SWE product variables involved in training. $x_i$ are the prediction feature variables, which contain two parts, one set contains the main feature variables of the gridded SWE products, and the other part

consists of the DEM auxiliary feature variables. $y_i$ is the observed snow water equivalent, and $\lambda$, $\beta$, $\beta_j$ and $\beta_0$ are the parameters to be solved. $1, \cdots, N$ is the sample of the training dataset. $\lambda \sum_{j=1}^{p} \beta_j^2$ is the penalty function terms.

**We added limitations of the RRM SWE product and directions for future improvement in the Results and Discussion sections.**

"Then, in addition to the DEM, meteorological elements, NDVI, land type, and other factors will affect the SWE estimation. Unfortunately, our current training model does not consider these factors in detail, which is a limitation of the current RRM SWE product."

■ The authors mentioned in the model training process, "... reduced the training data appropriately for the regions with denser training data, and make it close to the amount of training data in the sparse region" Please be more specific how this is done. Is it through randomly selecting training data in the regions with denser data? If so, what is the amount of the training samples used? Please provide more information.

**Reply: This suggestion has been accepted, and the text has been modified accordingly.**

For the region with dense training data, the training data were not simply selected randomly, but sample points that were spatially uniformly distributed were selected for the training as much as possible based on the latitude and longitude information of the observational points. The sample sizes of the training, validation and test datasets were divided according to the ratio of 7:2:1, where the numbers of training, validation and test set samples were 271651, 77614 and 38807, respectively.

**We have provided detailed supplementary explanations in Section 2.3.**

"The sample points that were spatially uniformly distributed were selected for training as much as possible based on the latitude and longitude information of the observational points."

"The total number of samples $N$ in the training dataset is 271651. The sample sizes of the training data set, validation data set and test data set are divided according to the ratio of 7:2:1, where the numbers of training set, validation set and test set samples are 271651, 77614 and 38807, respectively."

2. What is the range of the generated SWE product? Would be nice to see the range of generated SWE, and also the performances of SWE at different ranges. In Figure 4, it shows the data all below 400mm. What is the capability of this RRM product for capturing deeper snow? Please add more descriptions or discussions on this.

**Reply: This suggestion has been accepted, and the text has been modified.**

**Regarding the question of "**What is the range of the generated SWE product?**",** the range of the RRM SWE product is all land regions north of 45° N. The spatial range of the RRM SWE product is consistent with that of the DEM. In Figure 1, we labeled the range of the RRM SWE product in detail.

[Figure]

**Figure 1: The DEM and snow survey stations of the research region. The right subgraph shows the DEM, and the left subgraph shows the SWE observational stations. HSSC, hemispheric-scale snow course; RSSD, the Russian snow survey station. The spatial range of the RRM SWE product is consistent with that of DEM.).**

**Regarding the question of** "Would be nice to see the range of generated SWE, and the performances of SWE at different ranges," we evaluated the performances of the RRM SWE product in three representative regions: Russia, Canada, and Finland.

**We have made the following changes in the Abstract:**

"The average MAE and RMSE of the RRM SWE products are 0.22 and 19.92 mm at different altitude intervals and 0.21 and 27.00 mm at different regions, respectively."

**We have made the following changes in Section 2.4.2:**

"In addition, we also evaluated the performances of the RRM SWE product in three representative regions: Russia, Canada, and Finland."

**We have made the following changes in Section 3.2:**

**Table 3: Error list for the station data and RRM SWE product in different regions.**

| Region | MAE | RMSE (mm) | R | $R^2$ |
|--------|-----|-----------|---|-------|
| Russia | 0.20 | 26.39 | 0.89 | 0.79 |
| Canada | 0.23 | 29.31 | 0.87 | 0.76 |
| Finland | 0.21 | 25.29 | 0.89 | 0.79 |

"The RRM SWE product has good performance in different regions, and its RMSE in Russia, Canada, and Finland are 26.39 mm, 29.31 mm, and 25.29 mm, respectively; additionally, the performance of the RRM SWE product in different regions is basically similar (Table 3). The RRM SWE product not only performs well at different altitudes but also in different regions, and it has good stability."

**Regarding "In Figure 4, it shows the data all below 400mm. What is the capability of this RRM product for capturing deeper snow? Please add more descriptions or discussions on this,"** we retrained the model using hemispheric-scale snow course observational data to generate the new RRM SWE product. In Figure 4, the evaluation results for the SWE above 400 mm were added. The results show that the RRM SWE product still has a higher ability to capture the SWE above 400 mm than other products.

**We modified Figure 4 and added a discussion of the related content in the revised manuscript.**

[Figure]

**Figure 4: Error verification density diagram (a total of 38807 sample points were used for verification). The color bar represents the value of kernel density estimation. The closer the high-density area is to the 1:1 line, the higher the verification accuracy of the dataset is at most of the measuring stations.**

"For an SWE above 400 mm, the MAE and RMSE of the RRM SWE product and the measured SWE are 0.35 and 43.57 mm, respectively. Although the RRM SWE product is better than other products in capturing the SWE above 400 mm, it is still not as good at capturing the SWE below 400 mm relative to itself."

**Reply: Thank you for your suggestion. Figure 1 has been modified.**

[Figure]

**Figure 1: The DEM and snow survey stations of the research region. The right subgraph shows the DEM, and the left subgraph shows the SWE observational stations. HSSC, hemispheric-scale snow course; RSSD, the Russian snow survey station. The spatial range of the RRM SWE product is consistent with that of the DEM.**

**Reply: Thank you for your suggestion; the text has been modified accordingly.**

Because of systematic errors in the NOAA GHCN SWE data, we retrained, tested, and validated the model using the new hemispheric-scale snow course (HSSC) observational data and Russian snow survey data (RSSD) to generate a new version of the RRM SWE product and re-evaluated the product.

In addition, we compared the RRM SWE product with the SWE dataset obtained by the multisource data average method. The evaluation results are shown in Table 2 and Figure 3.

The new evaluation results show that the accuracy of the RRM SWE product is significantly improved compared to that of the ERA5-land SWE and GlobSnow SWE products. In addition, although the multisource data average method can improve the accuracy of SWE products to some extent (better than AMSR-E/AMSR2 SWE and GLDAS SWE), the improvement of this method is still very limited. The RRM SWE product has a significant advantage over the multisource data average method, and its accuracy is much higher than that of the simple multisource data average method.

**Regarding "**Additionally, it shows that GLDAS SWE has poor performance compared to other products. Have you considered to leave it out and further improve the machine learning model?**",** in preparing the RRM SWE product, the machine learning algorithm we constructed can automatically select the better-performing SWE training product based on the validation dataset. Therefore, although

the accuracy of the GLDAS SWE product is poor, it does not affect the training results of machine learning algorithms or the accuracy of the RRM SWE product.

**We have added specific content to the revised manuscript.**

"In addition, we compared the RRM SWE product with the SWE dataset obtained by the multisource data average method."

**Table 2: Error list for the station data and grid snow water equivalent products.**

| Error type | MAE | RMSE (mm) | R | $R^2$ |
|---|---|---|---|---|
| ERA-Interim | 0.43 | 46.81 | 0.69 | 0.48 |
| AMSR-E/AMSR2 | 0.49 | 52.39 | 0.47 | 0.22 |
| GLDAS | 0.58 | 65.25 | 0.52 | 0.27 |
| GlobSnow | 0.32 | 40.99 | 0.70 | 0.49 |
| ERA5-land | 0.32 | 37.02 | 0.84 | 0.71 |
| Multisource data average | 0.44 | 52.00 | 0.51 | 0.26 |
| RRM SWE | 0.21 | 25.37 | 0.89 | 0.79 |

[Figure]

**Figure 3: Accuracy comparison of various snow water equivalent products. The upper left sector represents the MAE, the upper right sector represents the RMSE, the lower-left sector represents R, and the lower right sector represents R². The sector axis represents the size of the error, and the color represents different SWE datasets.**

"Although the multisource data average method can improve the accuracy of SWE products to some extent (better than AMSR-E/AMSR2 SWE and GLDAS SWE), the improvement of this method is still very limited. The RRM SWE product has a significant advantage over the multisource data average method, and its accuracy is much higher than that of the simple multisource data average method."

---

## Author Response (AR2)

**Response to Topical Editor**

We greatly appreciate your valuable suggestions and comments. We studied your comments carefully and made substantial revisions accordingly. We hope that our corrections meet the requirements. We have also provided a clean version of the revised manuscript in which all revisions have been incorporated. The revised contents are marked in red in the manuscript. Our modifications and responses to the reviewers' comments are listed as follows. The reviewers' comments are in blue text, while the authors' responses are in black text. All revisions in the revised manuscript are in red text.

Comments to the author:

Dear Donghang Shao,

Thank you for a thorough revision of the manuscript and for addressing the reviewers' comments.

I have one main question that I will also ask separately to reviewer #1 (without starting a new review round):

I have quickly checked the GHCN and it seems that it contains direct SWE observations (such as from snow pillows at the SNOWTEL sites). So I do not understand why they are not suitable for the training of the model. You switched to the hemispheric scale snow course dataset but I do not understand why it is different, or better suited to your use, than the GHCN? Could you add maybe a sentence in the manuscript to clarify your choice of the HSSC?

**Reply: This suggestion has been accepted, and the text has been clarified and modified.**

GHCN applies pointwise observations on SWE (Menne et al., 2016), not distributed observations from snow courses. Therefore, GHCN SWE data cannot adequately reflect the spatial distribution characteristics of SWE or represent SWE at the spatial scale. The snow courses of hemispheric-scale snow course (HSSC) observational datasets are transects in which SWE is sampled manually at multiple locations with typical conditions to eliminate uncertainty in the regional-scale spatial variability of SWE due to the influence of snowpack characteristics and land cover type (Pulliainen et al., 2020).

Pulliainen et al.'s (Pulliainen et al., 2020) study of HSSC from Eurasia and North America showed that the typical exponential autocorrelation length of HSSC observations is 150~250 km. The HSSC dataset is representative of regional-scale SWE, and it is extremely suitable for comparison and validation with SWE satellite data or SWE reanalysis data.

References

Menne, M., Durre, I., Korzeniewski, B., McNeal, S., Thomas, K., Yin, X., Anthony, S., Ray, R., Vose, R., and Gleason, B.: Global Historical Climatology Network–Daily (GHCN-Daily), Version, 3, V5D21VHZ, 2016.

Pulliainen, J., Luojus, K., Derksen, C., Mudryk, L., Lemmetyinen, J., Salminen, M., Ikonen, J., Takala, M., Cohen, J., Smolander, T., and Norberg, J.: Patterns and trends of Northern Hemisphere snow mass from 1980 to 2018 (vol 41, pg 861, 2020), Nature, 582, E18-E18, 10.1038/s41586-020-2416-4, 2020.

**We added a detailed description of the reason for the selection of the HSSC data in Section 2.4.1.**

The snow courses of the HSSC dataset are transects in which SWE is sampled manually at multiple locations with typical conditions to eliminate uncertainty in the regional-scale spatial variability of SWE due to the influence of snowpack characteristics and land cover type (Pulliainen et al., 2020).

Additionally, your responses generally answer very well the reviewers' demands, but the quality of the English in the added material could be improved. I list some points below and I encourage a full proof-read of the manuscript before the next submission. The manuscript will be accepted once these minor comments are fixed.

**Reply: Thank you for your suggestion; the text has been modified.**

We have thoroughly checked and revised the English version of the article, including grammar problems and some sentences with unclear meanings, which makes the article more understandable. In addition, this manuscript has been revised by a native English speaker from American Journal Experts (https://www.aje.com/).

Sincerely,

Baptiste Vandecrux
* * *
(lines refer to the updated manuscript)

l. 23: "are 0.22 and 19.92 mm at different altitude intervals" This sentence is not clear. is it the average for different elevations? Is it the maximum MAE and RMSE among all elevation bins (if yes then which one?)?

**Reply: The values of 0.22 and 19.92 mm are the average values of MAE and RMSE for the RRM SWE product at different altitude gradients.**

"0.21 and 27.00 mm at different regions" Same thing here, are the two numbers MAE and RMSE or are they the min max values (of MAE or RMSE?) for different areas (then which regions?)?

**Reply: The values of 0.21 and 27.00 mm are the average values of MAE and RMSE for RRM products in Russia, Canada, and Finland.**

Consider using a structure like:
"The MAE (resp. RMSE) ranges from x1 mm (resp. y1 mm) for areas within E1-E2 m elevation to x2 (resp. y2) mm within the E3-E4 m elevation range. They are best in this region (spell out which one) and worst in that region (spell out)."

**Reply: This suggestion has been accepted, and the text has been modified.**

**We have modified the original sentence to:**

"The MAE ranges from 0.16 for areas within <100 m elevation to 0.29 within the 800-900 m elevation range. The MAE is best in the Russian region and worst in the Canadian region. The RMSE ranges from 4.71 mm for areas within <100 m elevation to 31.14 mm within the >1000 m elevation range. The RMSE is best in the Finland region and worst in the Canadian region."

l.175-176 "spatially uniformly distributed" The comment of the reviewer here is still not addressed: how do you select these points? how do you ensure that the selected samples are uniformly distributed? Please be thorough in your description so that someone that would like to reproduce your study would end up selecting (almost) the same samples for training.

**Reply: This suggestion has been accepted, and the text has been modified.**

**We added a detailed description of the training sample selection method in Section 2.3.**

"During the RRM model training process, we reconstructed the training data to try to extract training

samples that are uniformly distributed in space as much as possible. First, a scan window of 250 km × 250 km (10 × 10 pixels) was created. Then, each gridded SWE data point participating in training is scanned, and the sample numbers in each scan window are counted. Finally, the mean value $n$ of the sample numbers in all scan windows is taken as the number of training samples to be selected in each scan window. For the scan window with sample numbers higher than $n$, $n$ samples are randomly selected from the scan window. For the scan window with sample numbers lower than $n$, all samples in the scan window are selected as training samples."

l.261-265 Please refer to Table 2 in these two statements.

**Reply: This suggestion has been accepted, and the text has been modified.**

"The RRM SWE product has a significant advantage over the multisource data average method, and its accuracy is much higher than that of the simple multisource data average method (Table 2). Based on the above verification results, the accuracy of the RRM SWE is significantly improved; the RRM SWE dataset has higher accuracy than that of any single grid SWE dataset, and it also fills the gap in the original SWE data in terms of spatial and temporal resolutions."

l.278: I don't understand what you mean by "relative to itself"

Why is the RRM not so good below 400m? Do you mean that another product is better there? Then which one? Can you point at a graph or table that highlights a better product below 400 m? You can also merge this discussion point with your description of Figure 5.

**Reply: This suggestion has been accepted, and the text has been modified.**

We are very sorry for the misunderstanding of this sentence due to our lack of clarity. In fact, the meaning we intended to convey is as follows. The estimation accuracy of the RRM SWE product for the high value range of SWE (SWE > 400 mm) is lower than that for the low value range of SWE (SWE < 400 mm) (Fig. 4). The main reason for this is that the training accuracy of the RRM model for the high-value range of SWE is affected by the small number of stations that observe the high-value range of SWE.

**We have modified the original sentence to:**

"The estimation accuracy of the RRM SWE product for the high-value range of SWE (SWE > 400 mm) is lower than that for the low-value range of SWE (SWE < 400 mm) (Fig. 4). The main reason for this is that the training accuracy of the RRM model for the high-value range of SWE is affected by the small number of stations that observe the high-value range of SWE."

l. 284 "training model" change to "the RRM presented here". "in detail" change to "as predictor" or remove.

**Reply: This suggestion has been accepted, and the text has been modified.**

"Unfortunately, our current RRM presented here does not consider these factors as predictors, which is a limitation of the current RRM SWE product."

l.285 "remain challenging" Can you refer to a plot or a section in the study that would illustrate the poorer performance of the SWE products in rugged terrain? Maybe add a reference that deals with SWE estimation in rugged terrain.

**Reply: This suggestion has been accepted, and the text has been modified.**

"Finally, in complex terrain with an elevation interval >1000 m, the RRM SWE product performed poorly,

with an RMSE of 31.14 mm (Fig. 5), and the integration of SWE products remains challenging (Mortimer et al., 2020)."

References
Mortimer C, Mudryk L, Derksen C, Luojus K, Brown R, Kelly R, Tedesco M. 2020. Evaluation of long-term Northern Hemisphere snow water equivalent products. Cryosphere,14(5), 1579-1594. doi: 10.5194/tc-14-1579-2020

l.291-296: This >5 line long sentence is redundant with Figure 5. Consider shortening by only presenting the metrics or the elevation bins that are discussed in the text. The rest can be found in the Figure.

**Reply: This suggestion has been accepted, and the text has been modified.**

**We have reduced the original sentence to:**

"The above verification results show that the MAE, RMSE, R and $R^2$ between the RRM SWE product and measured SWE perform well at altitude gradients of <100 m, 100-200 m, 200-300 m, 300-400 m, 400-500 m, 500-600 m, 600-700 m, 700-800 m, 800-900 m, 900-1000 m and >1000 m (Fig. 5)."